# Enhancing Multi-tissue and Multi-scale Cell Nuclei Segmentation with Deep Metric Learning

**Tomas Iesmantas [1],[†], Agne Paulauskaite-Taraseviciene [2],[*],[†] and Kristina Sutiene [3],[†]**

[1]  Department of Applied Mathematics, Kaunas University of Technology, Studentu 50,
    51368 Kaunas, Lithuania; tomas.iesmantas@ktu.lt
[2]  Department of Applied Informatics, Kaunas University of Technology, Studentu 50, 51368 Kaunas, Lithuania
[3]  Department of Mathematical Modelling, Kaunas University of Technology, Studentu 50,
    51368 Kaunas, Lithuania; kristina.sutiene@ktu.lt
[*]  Correspondence: agne.paulauskaite-taraseviciene@ktu.lt
[†]  These authors contributed equally to this work.

**Abstract:** (1) Background: The segmentation of cell nuclei is an essential task in a wide range of biomedical studies and clinical practices. The full automation of this process remains a challenge due to intra- and internuclear variations across a wide range of tissue morphologies, differences in staining protocols and imaging procedures. (2) Methods: A deep learning model with metric embeddings such as contrastive loss and triplet loss with semi-hard negative mining is proposed in order to accurately segment cell nuclei in a diverse set of microscopy images. The effectiveness of the proposed model was tested on a large-scale multi-tissue collection of microscopy image sets. (3) Results: The use of deep metric learning increased the overall segmentation prediction by 3.12% in the average value of Dice similarity coefficients as compared to no metric learning. In particular, the largest gain was observed for segmenting cell nuclei in H&E -stained images when deep learning network and triplet loss with semi-hard negative mining were considered for the task. (4) Conclusion: We conclude that deep metric learning gives an additional boost to the overall learning process and consequently improves the segmentation performance. Notably, the improvement ranges approximately between 0.13% and 22.31% for different types of images in the terms of Dice coefficients when compared to no metric deep learning.

**Keywords:** nuclei detection; image segmentation; deep learning; metric embeddings; digital pathology

## 1. Introduction

The ongoing digitisation in pathology allows for computer-aided diagnosis in a wide spectrum of applications. In fact, the development of image recognition techniques enables the quantitative analysis of digital images with a high throughput processing rate. Automation continues to be widely applied in clinical practice since it both reduces the need for manual assessments, which are labor intensive and time consuming, while also reducing intra- and interobserver variability among pathologists [1,2]. Furthermore, automation can easily provide reproducible measurements and standardisation for follow-up evaluations and comparative studies used, for example, in personalised medicine. Quantitative information such as cell size, shape and spatial distribution are generally used by pathologists for cancer diagnosis [3]. For example, detecting nuclei enables the computer-aided assessment of immune cell infiltrates, which has a clear advantage due to its prognostic potential and use in immunotherapy trials [4,5]. Once accurately segmented, nuclear morphometric and appearance features can be beneficial for assessing cancer grades and also predicting treatment response [6]. Thus, a robust and generalised technique that accurately segments nuclei in a diverse range of images could be integrated into the computer-aided assessment technology used by pathologists.

The detection and segmentation of cell nuclei is one of the core operations in computer-aided diagnosis; it serves as a basis for cell counting and the study of subcellular morphology, such as investigations into the shape, size and textural properties [7]. This task is particularly challenging in histopathology image analysis, since the target cells, which are in many cases very small, are surrounded by background clutter made up of histological structures (capillaries, collagen, etc.) with irregular visual aspects, as well as artefacts which appear during image acquisition (see Figure 1). Second, significant variations in the appearance of cells are observed in different phases of mitosis development, which complicates the cell nuclei detection task. In particular, the amount of cells may range from tens to thousands in a typical high-resolution microscopy image, so the overlap of cells is a common phenomenon. Furthermore, the problem becomes more challenging because of the diversity of nuclear morphology, different staining conditions and different imaging technology applied. These challenges related to cell nuclei detection remain to be solved [2,5,8].

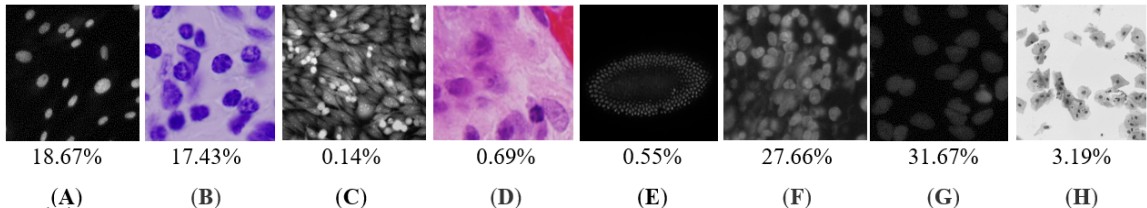

| 18.67% | 17.43% | 0.14% | 0.69% | 0.55% | 27.66% | 31.67% | 3.19% |
| (**A**) | (**B**) | (**C**) | (**D**) | (**E**) | (**F**) | (**G**) | (**H**) |

**Figure 1.** A diverse set of images and their proportions in the set: (**A**,**B**) Typical cases with a clear view; (**C**,**D**) rare instances surrounded by histological structures; (**E**) heavily clustered nuclei; (**F**) blurriness and a high level of overlapping; (**G**) an extremely high level of blurriness; and (**H**) an extremely high level of overlapping.

During the last few decades, many attempts have been made to improve the efficiency and accuracy of automatic nuclear segmentation techniques in digital pathology, with a special focus on histopathological image analysis. Recent comprehensive reviews of the nuclei segmentation literature [2,9] showed that image analysis techniques applied in this domain have evolved from classical approaches with hand-crafted features to deep learning-based techniques that typically yield superior accuracy. The most compelling advantage of deep learning is the ability to generalise and automatically learn problem-specific features directly from the original data. As such, deep convolutional neural networks (CNNs) demonstrated advancements in image recognition tasks and achieved state-of-the-art results in many medical imaging applications [8,10]. U-Net and DeepCell are examples of models that were designed specifically for cell or nuclei segmentation. Recently published results show that these architectures outperformed classical image processing algorithms in various segmentation tasks [11–14]. For example, the CNN architecture, called the comparison detector, is proposed for the detection of cervical cancer in small datasets [15]. In [16], a CNN-based algorithm is developed to perform cell segmentation from single channel images using different stain markers and magnifications. An original solution regarding how to achieve more accurate segmentation performance was provided in the paper [17], in which CNN was used to classify between the cell centre, cell body, and cell border. The Dice index, Jaccard coefficient and modified Hausdorff distance were used in the paper to evaluate the separation of overlapping cells.

Typically, most CNN architectures use cross-entropy loss function. Comparatively, discriminative metric learning loss functions have a better generalisation ability and are superior in solving few-shot learning problems. Therefore, they are highly likely to surpass the learning capabilities of traditional classification approaches for a small number of training samples. For example, in [18], a framework based on distance metric learning was proposed to classify mitotic $v/s$ interphase patterns. The authors relied on the metric learning methodology since the mitotic staining patterns are rather rare as compared with other interphase patterns. The distance metric computation was integrated with deep convolutional neural networks, which aimed to learn useful embeddings by distance comparisons of similar and dissimilar samples. Comparatively, in [19], a technique was presented to learn a distance

metric from the labeled data, which was locally adaptive to account for heterogeneity in the data. Additionally, a label propagation technique was used to improve the quality of the learned metric by expanding the training set using unlabelled data. Unlike other loss functions, such as mean square error loss or cross-entropy loss, metric learning aims to predict relative distances between samples. If the contrastive loss function is used, then the losses are computed by contrasting same-class or different-class pairs. Comparatively, the loss function computed using triplets of training data samples can be also used, which simultaneously increases the distances between the embeddings of the query and the negative sample (different samples) and at the same time reduces the distance between the positive and query samples (similar samples). The use of the above-mentioned loss functions in convolutional neural networks has proven to be very successful in various applications, such as face recognition [20], verification [21], fingerprint recognition [22], video separation [23] and clustering [24].

In this paper, we propose a deep learning-based model to segment cell nuclei from images. In fact, the developed model must be able to generalise across a wide range of tissue morphologies due to intra- and internuclear variations, where the overlap of cells is a common phenomenon. Furthermore, it must take into account the differences in staining protocols and imaging procedures. Therefore, we challenged the algorithm with a large-scale multi-tissue collection of microscopy image sets [25]. First, the algorithm was implemented using state-of-the-art CNN architectures, such as the dense convolutional network (DenseNet) and residual neural network (ResNet). Then, the metric embeddings were introduced in the model by adding additional terms into the loss function. Unfortunately, metric learning requires an a priori defined distance metric on the input space. In particular, standard metrics may ignore some important properties available in the dataset, thus resulting in the learning process being suboptimal. The search for a distance metric that brings similar data as close as possible (minimising intraclass variation), while moving nonsimilar data away (maximising interclass distance), can significantly increase the performance of these algorithms. To the best of our knowledge, this is the first attempt to segment cell nuclei via the deep metric learning technique.

## 2. Materials and Methods

We used image set `BBBC038v1`, available from the Broad Bioimage Benchmark Collection [25]. The dataset contains 670 images, which are highly varied because they came from many contexts: different organisms (humans, mice, flies), different nuclei treatments, different imaging conditions (fluorescent, histology stains), several magnifications and varying qualities of illumination, different states of nuclei including monolayers, tissues, embryos, cell division, genotoxic stress and differentiation (for example, see Figure 1). The dataset was constructed to challenge newly developed algorithms be able to generalise across these variations. As such, the dataset represents true variability, which is faced at biotechs, hospitals and by research groups at universities. Therefore, it was desirable to have a segmentation method with capacity enough to learn various aspects of each type of image in the dataset.

In the current study, two different state-of-the-art CNN architectures—ResNet and DenseNet—were implemented to build the customised deep neural network. These architectures included the typically used building blocks, but the composition of the layers was modified by introducing metric embeddings such as contrastive loss and triplet loss with semi-hard negative mining. In particular, we designed ResNet-68, ResNet-104 and DenseNet-71 architectures (see Section 2.1) and used them in the experimental study for multi-tissue cell nuclei segmentation. The implementation code can be found in the reference provided in the subsection "Supplementary Materials". It is important to note that the metric embedding was used as an addition to the cross-entropy and Dice losses. Therefore, in total, we implemented 9 models: 3 architectures × (2 metric embeddings + no metric embedding). Comparatively, in our recent study [26], we investigated a few more architectures, including DenseNet-27, DenseNet-71, ResNet-68, ResNet-104 and Vanilla UNet-54; however, some of them demonstrated a poor performance in segmenting the images, and therefore were not considered in the current paper.

Notably, some adjustments in the learning strategy were made in order to use the selected metric embeddings, i.e., contrastive loss and triplet loss with semi-hard negative mining. First, for each image, 200 pixels that included the nucleus and 200 pixels from the background were selected (400 pixels in total). This sampling was made randomly at each training iteration. Then, the predicted embeddings were extracted from the 12th network layer for those 400 pixels and fed into the metric learning loss function. Lastly, the results from the metric learning loss function were combined with the classical metrics used for the segmentation, such as Dice loss and cross-entropy loss. In other words, the final loss function $\mathcal{Loss}$ was constructed from three different subfunctions: metric learning loss $L_{MetricLearning}$ (which encourages clustering of pixels), Dice loss, and cross-entropy loss with equal weights, which is given by

$$\mathcal{Loss}\,(y,\hat{y},y_M,\hat{y}_M) = -\frac{1}{n\,N^2}\sum_{i=1}^{n}\sum_{j=1}^{N^2}\left[y_{ij}\log\hat{y}_{ij}+\left(1-y_{ij}\right)\log\left(1-\hat{y}_{ij}\right)\right]+$$

$$\frac{1}{n}\sum_{i=1}^{n}\left[1-\frac{2\sum_{j=1}^{N^2}\left(y_{ij}\,\hat{y}_{ij}\right)}{\sum_{j=1}^{N^2}y_{ij}+\sum_{j=1}^{N^2}\hat{y}_{ij}}\right]+L_{MetricLearning}\left(y_M,\hat{y}_M\right),$$

where $y$ is a true mask matrix with each column representing one mask of images (of size $N \times N$) reshaped into a column, in which there exist $n$ such columns for $n$ images in a single batch; $\hat{y}$ denotes the predicted segmentation for the entire batch; $y_M$ represents $M$ mask pixels, which were randomly selected to calculate the metric learning loss function value; $\hat{y}_M$ are embeddings of those $M$ pixels from the 12th layer.

Each architecture was trained with the Adam optimiser. The learning rate schedule was set to 100 epochs with a learning rate equal to 0.001 and 100 epochs with learning rate equal to 0.0005. In total, 200 epochs were enough to converge for each architecture.

### 2.1. Deep Learning Approach

The classical CNN architectures such as AlexNet, VGG and InceptionNet have a few layers stacked up on top of each other. In general, assumptions that deeper networks are better for image recognition and segmentation tasks are not absolutely true, because deeper networks become difficult to optimise, they encounter either vanishing or exploding gradient problems, as well as degradation problem (accuracy saturates and then degrades very fast). In order to overcome these problems, a novel architecture—ResNet—was proposed, which includes connections and adopted batch normalisation [27]. Actually, the architectures of the plain and residual networks are identical except for the skip connections added to each pair of filters. In particular, these connections do not require either an extra parameter or computational complexity. The main idea is to adjust the network so that it does not have to learn identity connections, but rather includes the inbuilt explicitly into the overall design. If new layers can be constructed as the identity mappings, a deeper model should have a training error no greater than its shallower parts. Therefore, the proposed ResNet architecture allowed for the development of even deeper networks (up to 152 layers) without compromising the model's accuracy and efficiency. These ultradeep architecture models with recurrent computations have shown excellent performance for recognition tasks.

To improve parameterisation and computational efficiency, the new architecture DenseNet [28], an extension of ResNet, was proposed. Similar to ResNet, DenseNet leverages skip connections between layers. In both ResNet and DenseNet, the input to a particular layer is formed by passing the output of all previous layers. However, different from Resnet, DenseNet outputs are concatenated rather than added. Furthermore, the layers in DenseNet do not receive a single output produced by the

summation as it is in Resnet, but all the outputs of the previous layers, which are concatenated in the depth dimension.

To facilitate the down-sampling in the architecture, DenseNet is composed of multiple dense blocks. DenseNet contains a convolutional layer, dense blocks, and transition layers between neighbouring dense blocks. The layers between these dense blocks perform convolution and pooling. Similar to ResNet, the number of convolutional layers used in each dense block can be optionally chosen. Furthermore, the features in DenseNet are used from all complexity levels, thus giving smoother decision boundaries. This also explains why DenseNet does not suffer from the underfitting problem and provides high performance results with a small training set. Compared with ResNet, DenseNet reduces the number of parameters to be estimated during the learning stage and at the same time can achieve better performance results with less complexity.

Considering all of this evidence, in the current study, we designed three CNN models: ResNet-104, ResNet-68, DenseNet-71 (see Table 1). Each of the architectures consisted of 13 building blocks. However, these blocks had different inner constructions and were iterated a different number of times depending on the architecture. The features from the 12th layer were fed into the contrastive loss or triplet loss with semi-hard negative mining. Then, the features from the last layer (13th) were used to calculate the value of Dice loss and cross-entropy loss. Finally, all three metrics were added together to give the final loss value. Both ResNet-104 and ResNet-68 consisted of blocks with an inner structure specified by a $1 \times 1$ convolution followed by a $3 \times 3$ convolution, and then again followed by a $1 \times 1$ convolution with the filters doubled. The results were then added to the previous layer using the simple identity function. DenseNet blocks consisted of $1 \times 1$ and $3 \times 3$ convolutions followed by the results being stacked on top of the features from the previous iteration. The same decoder structure was used for each of the architectures.

Image segmentation is usually treated as a pixelwise classification problem. In many cases, it is a binary classification with the label "0" being background and the label "1" indicating object pixels. Segmentation task (especially if considered within the deep learning framework) presupposes a metric (loss function) for checking how well the prediction from the deep artificial neural network agrees with the true segmentation. Over the years, many metrics have been proposed, but the most frequently used are cross-entropy loss, Dice loss, intersection over union loss, focal loss, and Tversky loss, as well as their combinations or generalisations [29].

Suppose $y$ refers to the true label, while $\hat{y}$ defines the prediction from the deep network. Then, the cross-entropy loss *CEL* function is defined as

$$CEL\left(y, \hat{y}\right) = -\left(y \log\left(\hat{y}\right) + \left(1 - y\right) \log\left(1 - \hat{y}\right)\right).$$

A generalisation of cross-entropy is a focal loss, which tries to down-weight the contribution of examples that are easy to predict, while ensuring more focus on harder examples [30]. As such, focal loss *FL* is given by

$$FL\left(y, \hat{y}\right) = -\left(\alpha\left(1 - \hat{y}\right)^{\gamma} y \log\left(\hat{y}\right) + \left(1 - \alpha\right) \hat{y}^{\gamma}\left(1 - y\right) \log\left(1 - \hat{y}\right)\right),$$

where $\gamma \geq 0$ is a focusing parameter that smoothly adjusts the rate at which easy examples are down-weighted; and $\alpha \in [0; 1]$ is a weighting factor. Note that the focal loss is equivalent to cross-entropy loss when $\gamma = 0$.

The two most frequently used measures of overlap—Dice loss *DL* and intersection over union loss *IoUL*—allow us to determine how accurate a predicted segmentation is, compared to a known/ground-truth segmentation. These loss functions are given by

$$DL\left(y, \hat{y}\right) = 1 - \frac{2y\hat{y} + 1}{y + \hat{y} + 1}$$

**Table 1.** Layers of convolutional neural network (CNN) architectures used for experiments.

| | Resnet-104 | | ResNet-68 | | DenseNet-71 | |
|---|---|---|---|---|---|---|
| Input layer: 256x256x1 grayscale image | | | | | | |
| L1 | $[3 \times 3, 16]$ | Conv1 | $[3 \times 3, 16]$ | Conv1 | $[3 \times 3, 16]$ | Conv1 |
| L2 | $\begin{bmatrix} 1 \times 1, 8 \\ 3 \times 3, 8 \\ 1 \times 1, 16 \end{bmatrix} \times 2$ | Conv2: 3*2=6 layers | $\begin{bmatrix} 1 \times 1, 8 \\ 3 \times 3, 8 \\ 1 \times 1, 16 \end{bmatrix} \times 1$ | Conv2: 3*1=3 layers | $\begin{bmatrix} 1 \times 1, 32 \\ 3 \times 3, 16 \end{bmatrix} \times 2$ | Conv2: 2*2=4 layers |
| Conv2+1 layer. Max pool $[1x1, 32]$ | | | | | | |
| L3 | $\begin{bmatrix} 1 \times 1, 16 \\ 3 \times 3, 16 \\ 1 \times 1, 32 \end{bmatrix} \times 2$ | Conv3: 3*2=6 layers | $\begin{bmatrix} 1 \times 1, 16 \\ 3 \times 3, 16 \\ 1 \times 1, 32 \end{bmatrix} \times 1$ | Conv3: 3*1=3 layers | $\begin{bmatrix} 1 \times 1, 32 \\ 3 \times 3, 16 \end{bmatrix} \times 2$ | Conv3: 2*2=4 layers |
| Conv3+1 layer. Max pool $[1x1, 64]$ | | | | | | |
| L4 | $\begin{bmatrix} 1 \times 1, 32 \\ 3 \times 3, 32 \\ 1 \times 1, 64 \end{bmatrix} \times 4$ | Conv4: 3*4=12 layers | $\begin{bmatrix} 1 \times 1, 32 \\ 3 \times 3, 32 \\ 1 \times 1, 64 \end{bmatrix} \times 2$ | Conv4: 3*2=6 layers | $\begin{bmatrix} 1 \times 1, 32 \\ 3 \times 3, 16 \end{bmatrix} \times 4$ | Conv4: 2*4=8 layers |
| Conv4+1 layer. Max pool $[1x1, 128]$ | | | | | | |
| L5 | $\begin{bmatrix} 1 \times 1, 64 \\ 3 \times 3, 64 \\ 1 \times 1, 128 \end{bmatrix} \times 4$ | Conv5: 3*4=12 layers | $\begin{bmatrix} 1 \times 1, 64 \\ 3 \times 3, 64 \\ 1 \times 1, 128 \end{bmatrix} \times 2$ | Conv5: 3*2=6 layers | $\begin{bmatrix} 1 \times 1, 32 \\ 3 \times 3, 16 \end{bmatrix} \times 4$ | Conv5: 2*4=8 layers |
| Conv5+1 layer. Max pool $[1x1, 256]$ | | | | | | |
| L6 | $\begin{bmatrix} 1 \times 1, 128 \\ 3 \times 3, 128 \\ 1 \times 1, 256 \end{bmatrix} \times 6$ | Conv6: 3*6=18 layers | $\begin{bmatrix} 1 \times 1, 128 \\ 3 \times 3, 128 \\ 1 \times 1, 256 \end{bmatrix} \times 3$ | Conv6: 3*3=9 layers | $\begin{bmatrix} 1 \times 1, 32 \\ 3 \times 3, 16 \end{bmatrix} \times 6$ | Conv6: 2*6=12 layers |
| Conv6+1 layer. Max pool $[1x1, 512]$ | | | | | | |
| L7 | $\begin{bmatrix} 1 \times 1, 256 \\ 3 \times 3, 256 \\ 1 \times 1, 512 \end{bmatrix} \times 6$ | Conv7: 3*6=18 layers | $\begin{bmatrix} 1 \times 1, 256 \\ 3 \times 3, 256 \\ 1 \times 1, 512 \end{bmatrix} \times 3$ | Conv7: 3*3=9 layers | $[3 \times 3, 512] \times 3$ | Conv7: 3 layers |
| Upsampling layers | | | | | | |
| L8 | Deconvolution$[2x2, 512]$. Add L6 output before max pooling $[3 \times 3, 256] \times 4$. 1+4 = 5 conv layers | | | | | |
| L9 | Deconvolution$[2x2, 256]$. Add L5 output before max pooling $[3 \times 3, 128] \times 4$. 1+4 = 5 conv layers. | | | | | |
| L10 | Deconvolution$[2x2, 128]$. Add L4 output before max pooling $[3 \times 3, 64] \times 4$. 1+4 = 5 conv layers | | | | | |
| L11 | Deconvolution$[2x2, 64]$. Add L3 output before max pooling $[3 \times 3, 32] \times 4$. 1+4 = 5 conv layers | | | | | |
| L12 | Deconvolution$[2x2, 32]$. Add L2 output before max pooling $[3 \times 3, 16] \times 4$. 1+4 = 5 conv layers. **Metrics are included**. | | | | | |
| L13 | $[1 \times 1, 1]$ Softmax. 1 convolutional layer | | | | | |
| $\Sigma$ | **104** | | **68** | | **71** | |

and

$$IoUL\,(y, \hat{y}) = 1 - \frac{y\hat{y} + 1}{y + \hat{y} - y\hat{y} + 1}.$$

The value of these two measures ranges between 0 and 1, with 0 indicating complete disagreement and 1 complete agreement between the true segmentation and the prediction.

Tversky loss *TL* is a generalisation of the Dice coefficient [31] given by

$$TL\,(y, \hat{y}) = 1 - \frac{y\hat{y} + 1}{y\hat{y} + \beta\,(1-y)\,\hat{y} + (1-\beta)\,y\,(1-\hat{y}) + 1},$$

where $\beta \in [0; 1]$ is a weighting factor.

Typically, the full network is trained according to a selected loss function. Different loss functions will produce different errors for the same prediction, and thus might change the performance of the model. On the other hand, every loss function has its own merits and pitfalls. Therefore, one might

ask the following question: is possible to also learn the metric while training the deep network? There is now extensive literature regarding this metric learning task. However, it was mostly formulated as a classification task, as it is understood in a classical way, i.e., prescribing a label for an entire image.

### 2.2. Deep Metric Embeddings

In its general form, the metric learning problem considers similarities between different pairs or triplets of samples coming from the dataset $X = (x_1, x_2, \ldots, x_N)$, where $x_i \in \mathcal{R}^n$ is $i$-th training sample and $N$ is the total number of samples in the dataset. As was reported in [32], these similarities are determined by the sets $S, D, R$ given by

$$S = \left\{ (x_i, x_j) \in X \times X : x_i \text{ and } x_j \text{ are similar} \right\},$$

$$D = \left\{ (x_i, x_j) \in X \times X : x_i \text{ and } x_j \text{ are not similar} \right\},$$

$$R = \left\{ (x_i, x_j, x_k) \in X \times X : x_i \text{ is more similar to } x_j \text{ than to } x_k \right\}.$$

The problem then consists of finding a distance $d \in \mathcal{D}$ that best fits the criteria specified by the similarity sets $S, D, R$. With respect to a certain loss function $\mathcal{L}$, the distance is found by optimising the problem

$$\min_{d \in \mathcal{D}} \mathcal{L}\left(d, S, D, R\right),$$

where $\mathcal{D}$ is a family of distances. Under the supervised learning setting with the true labels known $(y_1, y_2, \ldots, y_N)$, the similarity sets are reformulated as follows:

$$S = \left\{ (x_i, x_j) \in X \times X : y_i = y_j \right\}$$

$$D = \left\{ (x_i, x_j) \in X \times X : y_i \neq y_j \right\},$$

$$R = \left\{ (x_i, x_j, x_k) \in X \times X : y_i = y_j \neq y_k \right\}.$$

Loss function is one of the most important functions in deep feature learning. In general, the models presented in the the domain of deep metric learning can be categorised as minimising either the contrastive loss or the triplet loss [33,34]. If the contrastive loss $L_C$ [35] function is selected, the model is penalised differently based on whether the classes of the samples are the same or different. Specifically, if the classes agree, the loss function encourages the network to output feature vectors that are more similar, whereas if the classes differ, the loss function encourages the models to output feature vectors that are less similar. As such, contrastive loss $L_C$ is given by

$$L_C = \frac{1}{2}\left(1 - Y\right) D_W^2 + \frac{1}{2} Y \left\{ max(0, m - D_W) \right\}^2,$$

where $m$ is a margin value, which has to be selected manually; $Y$ indicates whether those two features come from the same class or not; $D_W(G(X_1), G(X_2))$ is a distance metric for a pair of input samples, where $X_1$ and $X_2$ is a pair of inputs in the training set, while $G(X_1)$ and $G(X_2)$ are two vectors generated as a new representation of paired samples in the embedding space. In the current paper, the distance metric $D_W$ is specified with a Euclidean metric for further analysis. Optionally, a more general metric, Mahalanobis distance, can be also considered. The function $G$ denotes the results of a next-to-last convolutional layer. For the case considered in this paper, this function refers to the results of a 12th layer.

Another metric considered in the current work is the triplet loss with semi-hard negative mining [20,34]. Instead of calculating the loss based on two examples, triplet loss $L_T$ involves three inputs: an anchor example $X$, similar to anchor input $X^S$ and dissimilar to anchor input $X^D$. As such, triplet loss $L_T$ is given by

$$L_T = \max\left(0, D_{W_1}^2 - D_{W_2}^2 + \alpha\right),$$

where $\alpha$ is the margin value; $D_{W_1} = D_W(G(X), G(X^S))$ and $D_{W_2} = D_W(G(X), G(X^D))$ are distance metrics for two vectors of new representations $G(X)$, $G(X^S)$ and $G(X^D))$, respectively. As one can see, triplet loss function penalises the model such that the distance between the matching examples is reduced and the distance between the nonmatching examples is increased.

## 3. Results

This section reports the validation results and main outcomes for all deep learning models considered in this paper. Table 2 compares the values of the area under the receiver operating characteristics (ROC) curve (area under the ROC curve (AUC)) obtained from a fivefold cross-validation performed on the dataset containing 670 images.

**Table 2.** Area under the receiver operating characteristics (ROC) curve (AUC): average $\pm$ standard deviation.

|  | DenseNet-71 | ResNet-68 | ResNet-104 |
|---|---|---|---|
| No metric learning | $0.97952 \pm 0.00198$ | $0.97176 \pm 0.00394$ | $0.98312 \pm 0.00272$ |
| Contrastive loss | $0.98703 \pm 0.00458$ | $0.98678 \pm 0.00351$ | $0.98873 \pm 0.00199$ |
| Triplet Loss | $0.98817 \pm 0.00225$ | $0.98902 \pm 0.00229$ | $0.99089 \pm 0.00150$ |

In Table 2, we can observe that in terms of mean and deviation of the AUC, the most promising result was achieved by the ResNet-104 architecture using triplet loss with semi-hard negative mining, while the poorest performance was demonstrated by the ResNet-68 architecture without metric learning. Surprisingly, the metric embeddings in the DenseNet architecture led to the increased variation of prediction performance over different folds compared to no metric learning. On the other hand, the overall AUC achieved over different folds is very high for each model. Therefore, it becomes very difficult or even impossible to improve this performance measure. Correspondingly, Table 3 reports the summary of Dice coefficient values achieved during the fivefold cross-validation experiment.

**Table 3.** Dice coefficient values: average $\pm$ standard deviation.

|  | DenseNet-71 | ResNet-68 | ResNet-104 |
|---|---|---|---|
| No metric learning | $0.8118 \pm 0.0043$ | $0.8109 \pm 0.0057$ | $0.8175 \pm 0.0098$ |
| Contrastive loss | $0.8233 \pm 0.0083$ | $0.8327 \pm 0.0048$ | $0.8401 \pm 0.0027$ |
| Triplet Loss | $0.8308 \pm 0.0065$ | $0.8357 \pm 0.0029$ | $0.8461 \pm 0.0034$ |

As shown in Table 3, the average values of Dice coefficient range from 0.8109 to 0.8461, with the highest result achieved by the Resnet-104 model using triplet loss with semi-hard negative mining. The performance demonstrated by the considered models corresponds with the results illustrated in Table 2. Here, we observe the same tendency, i.e., the use of metrics in the deep learning model enabled it to achieve a better segmentation performance compared to no metric learning. Notably, the variation among folds decreased when using ResNet metric learning, while DenseNet with embedded metrics demonstrated a larger deviation in the values of Dice coefficient. On the whole, cross-validation results give us assurance that the performance of a classifier is stable since the standard deviation is comparatively low.

In the following, we compare the segmentation prediction performance of the ResNet-104 architecture using triplet loss with semi-hard negative mining for testing the dataset consisting of 134 images). We relied on the Dice similarity coefficient, which is a preferable performance measure for the nuclei segmentation task, since it quantifies the similarity between the model output and reference mask. In particular, the Dice similarity coefficient was estimated separately for sample groups drawn up based on the type of image. Recall that the images in the dataset were acquired under a variety

of conditions and vary in cell type, cell spatial distribution, magnification, and imaging modality (brightfield vs. fluorescence). The obtained results are summarised in Figure 2.

| | Type of testing images | | | | |
|---|---|---|---|---|---|
| | | H&E stained | Papanicolaou (PAP) stained | Very dark with high level of blurriness | A huge amount of overlapping cells |
| All testing images | | ![H&E stained image] | ![PAP stained image] | ![Very dark image] | ![Overlapping cells image] |
| Dice coefficients (with metric learning) | | **0.84217** | **0.76490** | **0.5968** | **0.93788** | **0.84722** |
| Dice coefficients (without Metric learning) | | 0.81666 | 0.62536 | 0.57628 | 0.93670 | 0.84186 |

**Figure 2.** Dice coefficients estimated for one fold of testing samples based on the image type.

From the Figure 2, it is apparent that a 3.12% increase in the average value of Dice similarity coefficients with deep metric learning was achieved as compared to no metric learning. The largest gain of 22.31% is observed for the H&E-stained images, which is an appealing result since many authors found H&E-stained images challenging in terms of segmentation. Comparatively, a slight increase of 3.56% in the average value of Dice similarity coefficients is also seen for the PAP -stained images, while no significant improvement is observed for the other types of images included in the testing set. On the other hand, for dark images with a high level of blurriness or images with many overlapping cells, which seemed to us to be challenging for automated nuclei segmentation, the value of Dice coefficients achieved by the deep convolutional neural network is comparatively high, and metric embeddings had no significant effect on model performance. Figure 3 reports the difference in Dice coefficients for different types of images in the testing dataset, which was computed for every image individually.

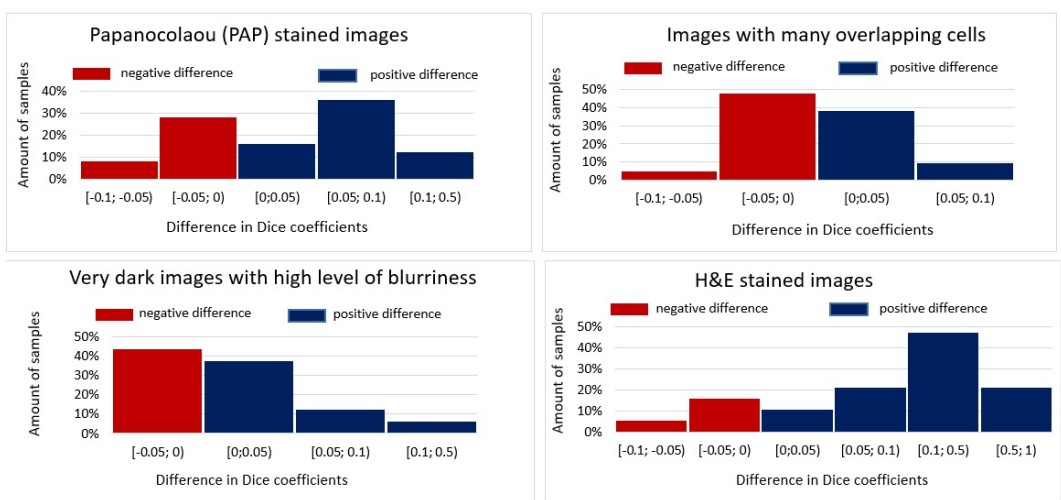

**Figure 3.** Differences in Dice coefficients achieved using the deep metric learning model and using the deep network model without metric learning for different types of images in the testing dataset.

In Figure 3, the positive differences indicate an improvement in prediction performance when deep learning with metrics was chosen, while the negative values show worsened performances. It can be seen that metric embeddings in the deep network model resulted in a similar amount of negative and positive Dice coefficients when images with many overlapping cells or very dark images with a high level of blurriness were under consideration (see Figure 3). This is in line with the average values of Dice coefficients reported in Figure 2, where the overall improvement was nearly zero, i.e.,

0.624% for images with many overlapping cells and 0.126% for dark images with a high level of blurriness. Notably, for the latter case, the improvement is slightly larger since we can observe more cases when the difference is larger than 0.1. A very similar situation can be observed for PAP-stained images. In fact, the most significant improvement is observed for H&E-stained images, where the Dice coefficients are in the range $[0.001; 1)$ for a major part of testing images. For this reason, we performed an in-depth analysis focusing solely on individual H&E-stained images in the testing set (see Figure 4). The figure plots the differences in Dice similarity coefficients achieved by the deep network model with metric learning (triplet loss semi-hard mining) and without metric learning.

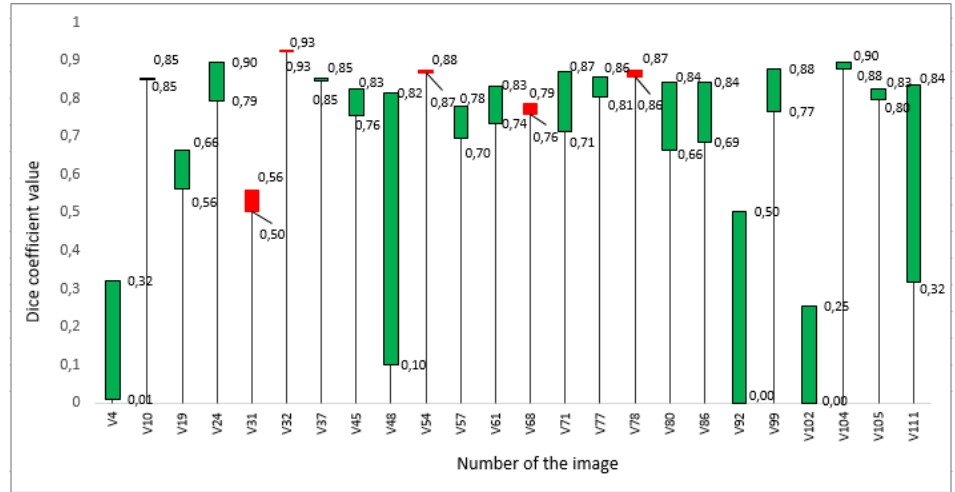

**Figure 4.** Dice coefficients estimated separately for individual H&E-stained images in the testing dataset: green bars show a gain in value, while red bars show a loss.

As one can see in Figure 4, the difference ranges from $-0.06$ to $0.72$. Overall, these results suggest that metric embeddings used in deep learning enhance cell nuclei segmentation applied to H&E-stained images. For demonstration purposes, Figure 5 depicts how the predictions differ when deep metric learning is used compared to when no metrics are embedded.

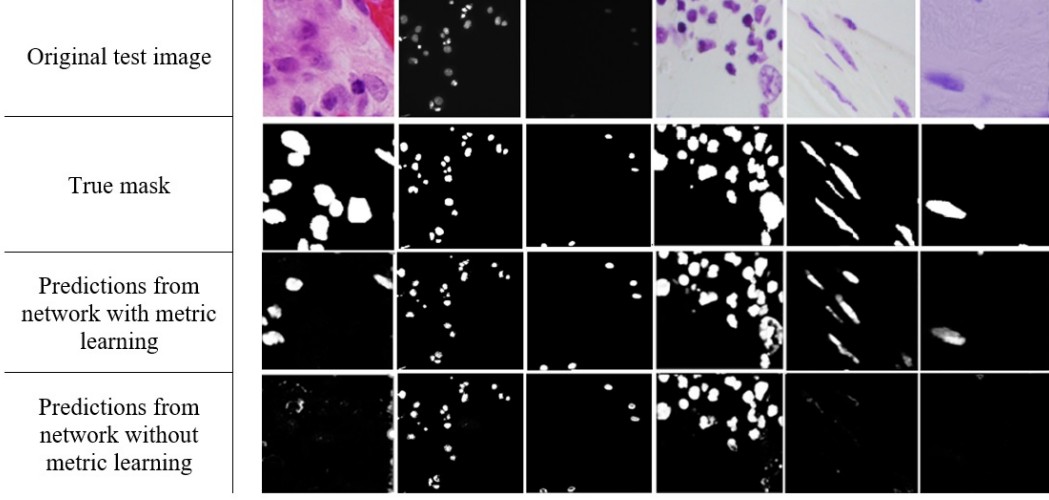

**Figure 5.** From top to bottom: original test image, its true mask, the predictions using the network with metric learning (triplet loss with semi-hard negative mining), and the predictions using the network without metric learning.

In Figure 5, the first column displays an example of an untypical image; these made up only 0.69% of the dataset. It can be seen that the embedded metrics produced predictions that are more

consistent with the true mask as compared to no metric learning. On the other hand, the relatively high accuracy achieved by both models and the nearly unobtrusive effect of embedded metrics are demonstrated in the second and third columns. This is in line with estimated Dice coefficients for dark images: 0.937 and 0.936, respectively (see Figure 2). The other three columns are provided to demonstrate the model performance for H&E-stained images, for which the use of metrics had the most profound effect in the experimental study. On the whole, one can observe that the use of metrics in the deep network model enhanced the segmentation prediction for images containing a complex background and higher variability. These observations hint at the possible use of metric learning for diverse sets of images containing different types of images.

To summarise, the best prediction results were obtained with the triplet semi-hard metric learning approach. The use of metric learning in addition to other loss functions (Dice and cross-entropy) resulted in a better representation of images. For demonstration purposes, the examples of this can be seen in Figure 6, where the edges are well differentiated and there is a clear distinction between the nucleus and the rest of the background when metric deep learning is considered (Figure 6, right), while without metric learning, this distinction is weak or almost nonexistent (Figure 6, left).

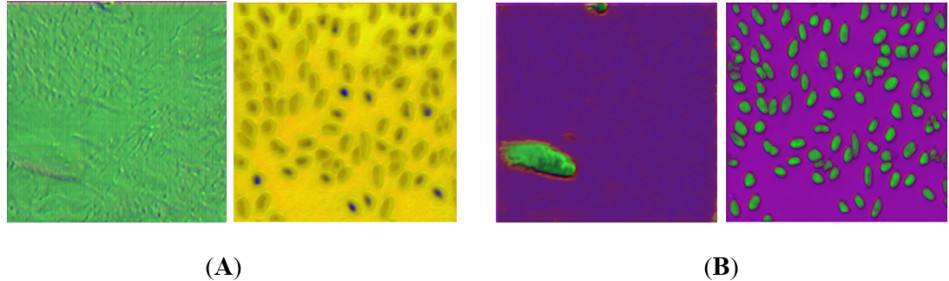

(A)　　　　　　　　　　　　　　　　　　　　　　　　(B)

**Figure 6.** (**A**) RGB images reconstructed from the layer before the last network layer ( i.e., from 16 feature maps) without metric learning; (**B**) RGB images reconstructed from the last network layer with metric learning (triplet loss with semi-hard mining). RGB channels were selected randomly from the 16 feature maps from the layer before the last layer of the network.

## 4. Discussion

The concept of deep metric learning was introduced when deep learning and metric learning were combined [34]. As such, the performance of a deep metric learning model depends not only on the quality of image dataset and the deep network architecture, but also on the metric loss function used in the learning. However, the selection of a metric is not always obvious, and no general rules exist regarding how the metric should be selected to guarantee the best performance of the algorithm. Therefore, in the current paper, we relied on two state-of-the-art deep network models, DenseNet and ResNet, embedded with metrics such as contrastive loss or triplet loss with semi-hard negative mining selected optionally.

While testing the segmentation prediction on the testing dataset using Dice similarity coefficients, it was observed that the metric embeddings used in the deep network model allowed us to significantly improve the cell nuclei segmentation in H&E-stained images within the diverse set of microscopy images. It is well known that the visual inspection of H&E-stained images is still a challenging task and often leads to the over-segmentation phenomenon, because of large variation among H&E-stained images, uneven intensities of the background regions and heterogeneity of cell nuclei [36]. Furthermore, cell nuclei segmentation, being such as important task in cancer detection, is complicated because of cells which often touch and overlap, making the separation problem of the cell nuclei difficult [37,38]. Comparatively, many researchers demonstrated successful segmentation results of nuclei in H&E-stained images by manipulating colour spaces in the image [39] and then using certain thresholds [10]. However, most of these algorithms are applicable only for the separation of nuclei, which have an ellipse-like shape [40]. In general, we infer that the mean value 0.764 of Dice similarity

coefficient achieved in the current paper for the large-scale multi-tissue collection of images is in line with already published studies, where its value varied from 0.69 to 0.9 when the deep learning model was applied [6,41,42]. However, the comparison of the obtained results is not trivial because of the different algorithms applied and the accuracy metrics selected, as well as the different image qualities, magnifications ($40\times$ or $20\times$) and different staining techniques, etc., in the dataset.

Future work will concentrate on a quantification of cell nuclei in complex tissue images, since numerous procedures in medical and biological investigations require the counting of cells [43]. The research will be continued using the same image set BBBC038v1 that was used in the current paper. For comparison, the authors in [44] addressed the question of cell nuclei counting, considering the same image set as we did. For this purpose, they proposed a technique to automate the quantification of cell nuclei in histological images. However, the proposed approach failed to count nuclei correctly for complex images, such as those with no dominant background or that contain large blue or red areas. Therefore, we believe that we can continue with the use of deep metric learning for the quantification of histological images.

## 5. Conclusions

In this paper, we proposed a deep metric learning-based model to segment cell nuclei from images. The model was designed by customising two deep network models—the DenseNet and ResNet architectures. Adjustments in the learning strategy were made in order to use the metric embeddings, by which either contrastive loss or triplet loss with semi-hard negative mining can be selected.

One of our aims was to explore how the developed model is able to generalise across a wide range of tissue morphologies due to intra- and internuclear variations, where the overlap of cells is a common phenomenon, as well taking into account the differences in staining protocols and imaging procedures. During the experimental investigation, we observed a 3.12% increase in the average value of Dice similarity coefficients using deep metric learning as compared with no metric learning; with the largest gain of 22.31% being observed for H&E-stained images when the ResNet-104 model was embedded with triplet loss with semi-hard negative mining. In fact, the ResNet-68 architecture demonstrated the poorest performance, while metric embeddings in the DenseNet architecture increased the variation in the performance over different cross-validation folds compared to no metric learning.

There is a good reason why including metric learning boosts the accuracy of deep learning methods, i.e., metric learning gives an additional flexibility allowing the model to learn better representations. Predetermined loss functions alone (dice loss, cross-entropy, etc.) may not be optimal for a given problem, while enabling the possibility to learn better distance metrics provides the possibility of better data embeddings, which results in higher accuracy. In this paper, the pixels selected for metric learning were chosen randomly. However, this strategy could be possibly improved using some heuristic selection procedures, such as choosing poorly embedded pixels or selecting only a fraction of a nucleus from the sample pixels. This shall be investigated in more detail in the near future. Given the fact that images in most biomedical applications have very high variability, and that large quantities of images for most problems are not available, metric learning may provide a chance for higher accuracies with smaller samples of images. As such, we believe that our research contributes to future studies in the field of biomedical science.

**Supplementary Materials:** The code developed during this study is available at http://www.mdpi.com/2076-3417/10/2/615/s1.

**Author Contributions:** Conceptualisation, A.P.-T. and K.S.; Methodology, A.P.-T., K.S. and T.I.; Software T.I.; Validation, A.P.-T. and T.I.; Visualisation, A.P.-T. and K.S.; All authors assisted in writing and improving the paper. All authors have read and agreed to the published version of the manuscript.

**Funding:** This research was funded by project "R&D of Cell Nucleus Detection Model Based on Artificial Intelligence", Kaunas University of Technology, grant number PP-91G/19.

**Conflicts of Interest:** The authors declare no conflict of interest.

## Abbreviations

The following abbreviations are used in this manuscript:

| | |
|---|---|
| AUC | Area under the ROC curve |
| CNN | Convolutional Neural Network |
| DenseNet | Dense Convolutional Network |
| ResNet | Residual Neural Network |
| ROC | Receiver Operating Characteristics |

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
