# Peer review of "Enhancing Multi-tissue and Multi-scale Cell Nuclei Segmentation with Deep Metric Learning"

_applsci, doi:10.3390/app10020615_

Round 1

Reviewer 1 Report

This paper presents a deep learning method for Cell Nuclei Segmentation with a new loss term. The Cell Nuclei Segmentation is a very challenging task, and the method indeed can improve segmentation accuracy.

Comment-1:

"Typically, most of CNN architectures use adopted cross-entropy based softmax loss function"

Change " adopted cross-entropy based softmax loss " to "cross-entropy loss" which is enough

Comment-2:

Please add the reference papers for " focal loss" and " Tversky loss "

Comment-3:

"DW(G(X1), G(X2)) is some distance metric for a pair of input"

Please explain the distance metric. Is it Euclidian distance or something else?

Comment-4:

"while G(X1) and G(X2) are two vectors generated as a new representation of paired samples in embedding space."

Please explain G(X1): where does G(X1) come from?   Some layer of the network? I assume it refers to "embeddings of those M pixels from the 12th layer."

Comment-5:

Please give more information about training, validation and testing sets, .i.e. how many images per set.

" 5-fold cross-validation"- is it performed on training set ?

Comment-6:

About Figure 2, why not reports DICE of all five cross validation sets like Table 2?

The link does not work

" Supplementary Materials: The code developed during this study is available at [name of repository] https://www.mdpi.com/2076-3417/xx/1/5/s1."

Author Response

Please, find the response letter attached.

Reviewer 2 Report

Authors present a novel segmentation technique for nuclei segmentation. They have studied the performance of CNN using different metrics. Their overall system achieves an accuracy of 0.13% and 22.31%. However, I would recommend them to study the performance U-Net and add the same in the comparison table. Below is a paper on U-Net segmentation used for X-ray segmentation. Please study the performance of the same and see how the results. U-Net segmentation have proven to be highly effective for Bio medical imaging applications. 

- https://www.researchgate.net/publication/332589403_A_Computationally_Efficient_U-Net_Architecture_for_Lung_Segmentation_in_Chest_Radiographs

Author Response

(The authors gave the same response as above.)

Round 2

Reviewer 1 Report

thanks for answering my questions